# Pheromone components affect motivation and induce persistent modulation of associative learning and memory in honey bees

David Baracchi[1,2✉], Amélie Cabirol[3], Jean-Marc Devaud[1], Albrecht Haase[3,4], Patrizia d'Ettorre[1,5,6,9] & Martin Giurfa [1,7,8,9✉]

Since their discovery in insects, pheromones are considered as ubiquitous and stereotyped chemical messengers acting in intraspecific animal communication. Here we studied the effect of pheromones in a different context as we investigated their capacity to induce persistent modulations of associative learning and memory. We used honey bees, *Apis mellifera*, and combined olfactory conditioning and pheromone preexposure with disruption of neural activity and two-photon imaging of olfactory brain circuits, to characterize the effect of pheromones on olfactory learning and memory. Geraniol, an attractive pheromone component, and 2-heptanone, an aversive pheromone, improved and impaired, respectively, olfactory learning and memory via a durable modulation of appetitive motivation, which left odor processing unaffected. Consistently, interfering with aminergic circuits mediating appetitive motivation rescued or diminished the cognitive effects induced by pheromone components. We thus show that these chemical messengers act as important modulators of motivational processes and influence thereby animal cognition.

[1] Research Centre on Animal Cognition, Center for Integrative Biology, CNRS, University of Toulouse, 118 route de Narbonne, F-31062 Toulouse, Cedex 09, France. [2] Department of Biology, University of Florence, Via Madonna del Piano, 6, 50019 Sesto Fiorentino, Italy. [3] Center for Mind/Brain Sciences (CIMeC), University of Trento, Piazza Manifattura 1, I-38068 Rovereto, Italy. [4] Department of Physics, University of Trento, Via Sommarive 14, I-38123 Povo, Italy. [5] Laboratory of Experimental and Comparative Ethology, University of Paris 13, F-93430 Sorbonne Paris Cité, France. [6] Institut Universitaire de France (IUF), Paris, France. [7] Institut Universitaire de France (IUF), Toulouse, France. [8] College of Animal Science (College of Bee Science), Fujian Agriculture and Forestry University, Fuzhou 350002, China. [9]These authors contributed equally: Patrizia d'Ettorre, Martin Giurfa. ✉email: david.baracchi@unifi.it; martin.giurfa@univ-tlse3.fr

Pheromones are chemical signals released by a sender into the environment to convey specific messages to members of the same species, in which they release stereotyped and adaptive responses[1]. Since their definition, proposed six decades ago, pheromones have been confined to an intraspecific communication scenario, so that less is known about their possible consequences in contexts different from that of information exchange[2]. In particular, if exposure to pheromones affects in a durable way subsequent associative learning and memory remains unknown. Research on rodents[3,4] and rabbits[5] has shown that some pheromone components can replace reinforcing stimuli *during* associative conditioning and mediate the learning of odor or contextual cues. In contrast to these works, we asked if *prior* exposure to pheromone components induces subsequent and persistent motivational changes affecting the way in which animals learn and memorize when the pheromone signals are no longer present. Motivation is central to animal and human behavior as it affects decision-making processes, stimulus searching or avoidance, and consequently, what individuals learn and remember[6–8]. If besides providing specific messages to conspecifics, pheromones change an animal's motivation according to their message, they will exert important consequences on its capacity to learn and memorize.

A relevant species to address this hypothesis is the domestic honey bee *Apis mellifera*, which represents a pinnacle of sociality among insects[9]. Social cohesion and colony efficiency rely largely on pheromones, which signal diverse events in a variety of behavioral contexts[10,11]. Furthermore, individual worker bees exhibit impressive learning and memory abilities[12,13], which can be studied in the laboratory using controlled conditioning protocols. One of these protocols is the olfactory conditioning of the proboscis extension reflex (PER), in which harnessed bees learn the association between an odorant (the conditioned stimulus or CS) and a reward of sucrose solution (the unconditioned stimulus or US), so that they exhibit the appetitive PER to the odorant that anticipates the food[14,15]. We took advantage of this protocol to determine if prior pheromone exposure modifies subsequent associative learning and memory when pheromone components were no longer present. To disentangle the effect of these components on memory formation and retrieval, which may engage different processes and neural circuits[16], we exposed honey bees either before conditioning, thereby affecting ongoing learning and memory formation, or after learning and before a retention test, affecting exclusively the process of retrieval. We studied the impact of two pheromonal signals of different "valence": 2-heptanone (2H), a deterrent pheromone-signaling aversive situation[10,17], and geraniol (GER), the major component of the attractive pheromone of the Nasonov gland[10,18]. We preexposed bees to these pheromone components to determine if they affect subsequent olfactory learning and memory formation. Using in vivo two-photon calcium imaging, we analyzed the impact of this preexposure on odor-evoked neuronal activity in the antennal lobes (ALs), the primary olfactory centers of the insect brain. In parallel, we analyzed how these pheromonal signals affect aminergic circuits mediating appetitive motivation and responsiveness in the bee brain[19–21]. Our results show that GER and 2H do not modify olfactory processing, but induce long-lasting changes in motivation, which affect subsequent learning and memory formation. Thus, besides acting as chemical messengers, pheromonal signals contribute to behavioral plasticity by preparing individuals to learn and memorize according to pheromonal valence.

## Results

### Modulation of olfactory learning and memory by pheromones.
We first studied if pheromone components modify learning performances using the olfactory conditioning of the PER[14,15]. We preexposed bees to either Geraniol (GER), an attractive pheromone component[18], or 2-Heptanone (2H), a deterrent pheromone-signaling aversive event[17]—during 15 min and waited for additional 15 min before starting conditioning (Fig. 1a). Control bees were preexposed to mineral oil, a standard control solvent in studies on olfaction, which was used to dilute pheromone components. During conditioning and in subsequent tests, an air extractor ensured that no pheromone leftovers remained at the experimental site. The three groups of bees were conditioned to discriminate the floral odors limonene and eugenol in the absence of pheromonal stimulation during ten trials spaced by 12 min. Thus, 123 min elapsed since pheromone-component exposure and the end of conditioning. Conditioning consisted of five rewarded presentations of one of the two odorants with sucrose solution (CS+) and five unrewarded presentations of the other odorant (CS−). Within each group, the role of limonene and eugenol as CS+ and CS− was counterbalanced, and the sequence of odorants was pseudorandomized[22].

Bees in all three groups learned to discriminate the CS+ from the CS− (generalized linear mixed model, GLMM, trial: $\chi^2 = 70.93$, df $= 1$, $p < 0.0001$) and responded differently to both odorants at the end of training (GLMM, CS: $\chi^2 = 44.87$, df $= 1$, $p < 0.0001$) (Fig. 1b). Bees exposed to GER performed better than controls, while the opposite occurred in bees exposed to 2H (treatment × CS interaction: $\chi^2 = 30.48$, df $= 2$, $p < 0.0001$). Differences were based on responses to the CS+ as GER-exposed bees responded more to the CS+ than control bees (GLMM, Tukey's post hoc test; CS+: $p = 0.0007$; CS−: $p = 0.80$), while 2H-exposed bees responded less to the CS+ than controls ($p = 0.018$). No differences were found in the case of CS− responses ($p = 0.98$). Thus, preexposure to GER facilitated subsequent appetitive olfactory learning, while preexposure to 2H impaired it. Similar results were obtained upon evaluation of individual acquisition scores (ACQS), calculated as the sum of responses to the five CS+ presentations for every preexposed bee (see Supplementary Fig. 1).

The same three groups previously preexposed and conditioned were further tested with the CS+ and the CS− 2, 24, and 72 h after conditioning (Fig. 1a). In this way, besides evaluating how preexosure to pheromonal components affected their learning, we evaluated how it affected memory retention after conditioning. As a robust proxy of memory retention, we quantified the percentage of bees exhibiting specific memory, i.e., responding only to the CS+ and not to the CS−[22,23]. Preexposure to pheromone components of different valence induced opposite modulation of specific memory (Fig. 1c, GLMM, treatment: $\chi^2 = 27.80$, df $= 2$, $p < 0.0001$). The time elapsed between training and test also affected specific memory (testing time: $\chi^2 = 6.03$, df $= 2$, $p < 0.049$). The interaction was not significant ($\chi^2 = 6.69$, df $= 4$, $p = 0.153$). Overall, a lower percentage of bees with specific memory was observed after 2H preexposure with respect to controls (Tukey's post hoc test: $p < 0.001$), while a higher proportion was observed after GER preexposure (Tukey's post hoc test: $p < 0.001$).

We next asked if preexposure to GER and 2H affects memory retrieval, irrespective of the differences in learning induced by preexposure. We trained unexposed bees to discriminate limonene and eugenol, and selected the bees exhibiting correct discrimination (responding to the CS+ and not to the CS−) in the last conditioning trial. Thus, all bees had the same acquisition level at the end of training. They were then exposed to GER, 2H, or mineral oil during 15 min, prior to the memory tests performed 2, 24, or 72 h after conditioning (Supplementary Fig. 2A). Different groups of bees were used for each treatment and test. Retrieval performances were similar between groups at

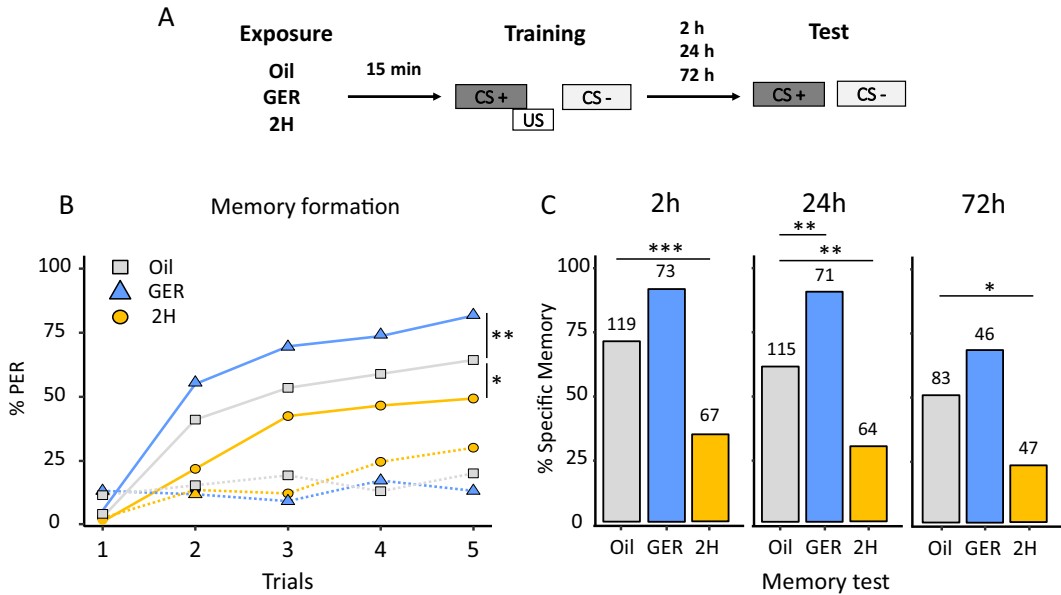

**Fig. 1 Pheromone components modulate associative olfactory learning and memory retention in honey bees. a–c** Pheromone components modulate associative olfactory learning and memory according to their valence. **a** Experimental protocol used. **b** Associative olfactory conditioning of PER in honey bees exposed to geraniol (GER), 2-heptanone (2H), or mineral oil 15 min before training. Proportion of bees showing a conditioned response (PER) to the rewarded (solid lines) and unrewarded (dotted lines) odorants during successive conditioning trials. GER-preexposed bees ($n = 75$ independent bees) performed better than control bees exposed to mineral oil ($n = 129$ independent bees), while 2H preexposed bees ($n = 73$ independent bees) performed worse than controls. (*) $p < 0.05$; (**) $p < 0.001$. **c** Retention tests performed 2, 24, or 72 h post conditioning. Preexposure to GER (blue), 2H (orange), or mineral oil (gray) was performed prior to conditioning. The figure shows the proportions of bees showing a specific memory (i.e., responding to the CS+ and not to the CS−) in the retention tests. The sample size is reported above each bar and refers to independent bees. Retention was better in bees exposed to GER compared with control bees. The opposite was observed in bees exposed to 2H. (*) $p < 0.05$; (**) $p < 0.01$; (***) $p < 0.001$.

any testing time, and irrespective of the pheromone component exposed (Supplementary Fig. 2B, logistic regression, 2 h: treatment: $\chi^2 = 0.27$, $df = 2$, $p = 0.9$, 24 h: treatment: $\chi^2 = 0.31$, $df = 2$, $p = 0.9$, 72 h: treatment: $\chi^2 = 1.48$, $df = 2$, $p = 0.5$). Thus, preexposure to pheromone components differing in valence did not affect memory retrieval.

As olfactory PER conditioning is a case of Pavlovian learning involving repeated exposures to an odorant and sucrose solution[14], we next studied the effect of GER and 2H preexposure on the processing of odorants and on appetitive motivation evaluated through sucrose responsiveness.

**Effect of pheromone components on odor processing.** We first analyzed the effect of GER and 2H preexposure on odorant processing by recording odor-evoked neural activity in the ALs, the primary olfactory center in the bee brain, following pheromone-component preexposure. ALs are constituted by glomeruli, which are interaction sites between afferent olfactory receptor neurons located on the antennae, local interneurons, and projection neurons (PNs) conveying the olfactory information to higher-order brain centers[24]. Odorants are encoded in the ALs as odor-specific glomerular maps, which can be visualized using in vivo calcium imaging[25]. Using a fluorescent calcium-sensitive dye, we recorded PN activity at the level of the ALs by means of two-photon fluorescence microscopy. Bees prepared for imaging were preexposed to GER, 2H, or mineral oil. PN responses to limonene and eugenol, the conditioned odorants, were recorded 15 min before pheromone exposure (baseline), as well as 15 min and 2 h after preexposure, in the absence of training (Fig. 2a). The latter recording time corresponds to the end of conditioning in trained animals, when odorant discrimination was achieved. In this way, we determined if pheromones affected per se (i.e., in the absence of training) perceptual

distances between odorants, thus modulating their discrimination. Fluorescence was recorded along a scanline crossing multiple glomeruli (Fig. 2b, c) upon alternate presentations of limonene and eugenol, lasting 4 s each. The normalized fluorescence-intensity change ($-\Delta F/F$) provides a measure of the neuronal firing rate (Fig. 2d).

Preexposure to mineral oil, GER, and 2H affected neither limonene nor eugenol encoding in the ALs. The neural activation averaged over all glomeruli in response to odor stimulation was not different along recording times after exposure to pheromone components or mineral oil (Fig. 2e; Friedman test, $GER_{Limonene}$: $\chi^2 = 1.75$, $df = 2$, $p = 0.42$; $GER_{Eugenol}$: $\chi^2 = 3.25$, $df = 2$, $p = 0.2$; $2H_{Limonene}$: $\chi^2 = 1.75$, $df = 2$, $p = 0.42$; $2H_{Eugenol}$: $\chi^2 = 5.25$, $df = 2$, $p = 0.07$; Mineral $Oil_{Limonene}$: $\chi^2 = 2.25$, $df = 2$, $p = 0.32$; Mineral $Oil_{Eugenol}$: $\chi^2 = 3.25$, $df = 2$, $p = 0.2$). Similarly, the Euclidian distance (ED) between the glomerular-activation patterns of limonene and eugenol, which constitutes a measure of their discriminability[26,27], did not change following pheromone-component or mineral-oil preexposure (Fig. 2f; Friedman test, GER: $\chi^2 = 2.25$, $df = 2$, $p = 0.32$; 2H: $\chi^2 = 3.25$, $df = 2$, $p = 0.2$; Mineral Oil: $\chi^2 = 4$, $df = 2$, $p = 0.14$). Thus, the modulation of odor learning and memory induced by pheromone components does not occur via olfactory (CS) circuits.

**Effect of pheromone components on appetitive motivation.** We reasoned that pheromonal signals might affect appetitive motivation, i.e., the subjective evaluation of sucrose reward, thereby determining different levels of appetitive learning (Fig. 1b). To test this hypothesis, we studied if preexposure to GER and 2H induces opposite modulations of sucrose responsiveness, using a standard test in which bees are stimulated with six increasing concentrations of sucrose solution delivered to the antennae (0.1, 0.3, 1, 3, 10, and 30% w/w)[28,29].

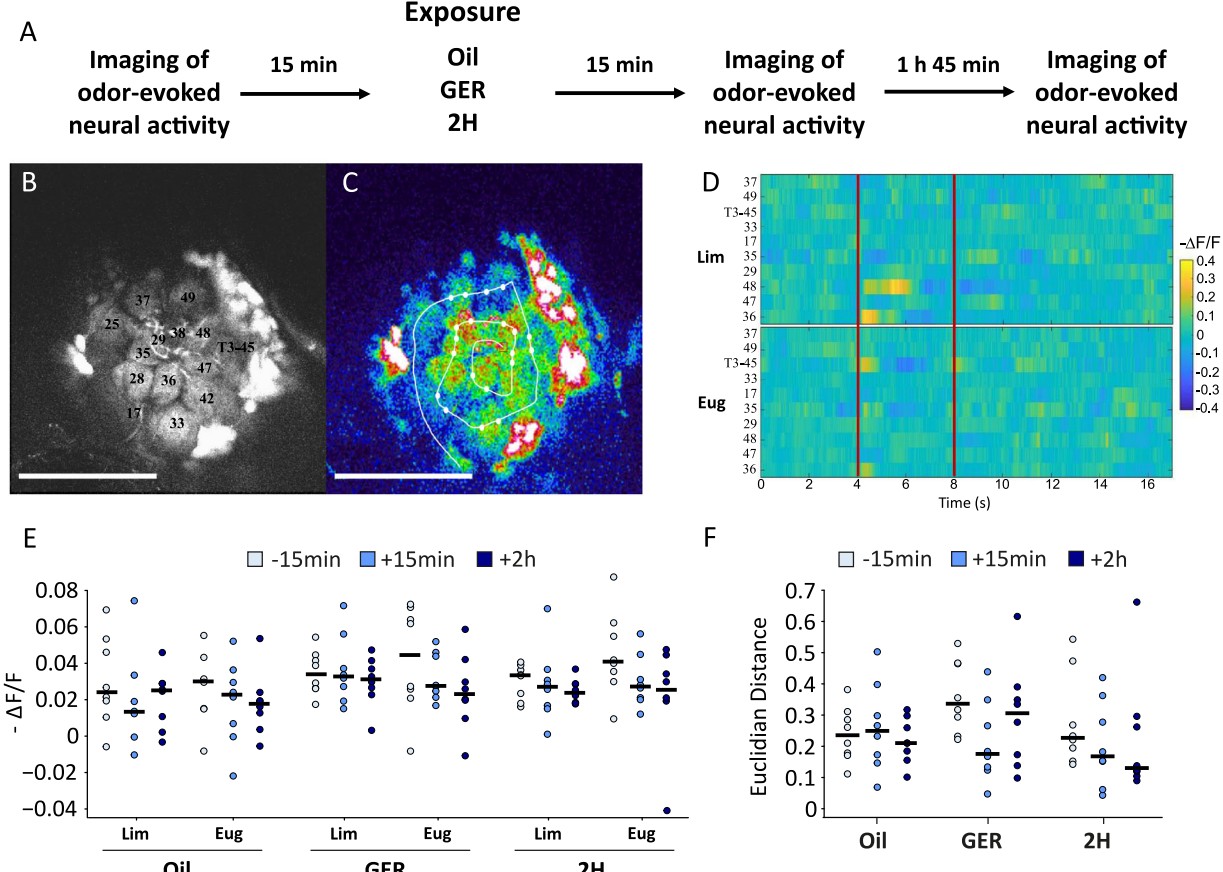

**Fig. 2 Effect of pheromone components on neural odor processing. a–f** Pheromone components do not affect olfactory coding and odorant similarity in the antennal lobe. **a** Experimental protocol used. **b** Two-photon microscopy image of the left antennal lobe (AL) stained with a fluorescent calcium-sensitive dye injected in the projection neurons (PN) of the medial and the lateral antennal protocerebral tracts. Activity in PN dendrites in AL glomeruli can be visualized in this way. **c** Signal intensity was recorded along a line crossing ten glomeruli identified using the antennal lobe atlas of the honey bee[62] (AN antennal nerve, v ventral, l lateral, m medial, d dorsal. Numbers refer to identified glomeruli) and averaged between the identified borders of each glomerulus (white circles). Scale bars = 200 μm. **d** Signal intensity of each identified glomerulus over time after background subtraction and normalization ($-\Delta F/F$). Red lines represent the onset and offset of limonene (Lim, above) and eugenol (Eug, below) presentation to the bee. **e** Change in normalized fluorescence during the first 600 ms of odor stimulation with Lim and Eug 15 min before ("−15 min") mineral oil ("Oil"; $n = 8$ independent bees) or pheromone- component (GER, 2H; $n = 8$ independent bees for both components) exposure, as well as 15 min ("+15 min") and 2 h ("+2 h") after exposure. No significant variation in activity was found for each odorant over time for both pheromonal treatments. **f** Euclidian distance—a measure of odor distinguishability—in the odor-coding space defined by the activity recorded for the ten identified glomeruli between the odor representations of Lim and Eug 15 min before, as well as 15 min and 2 h after the exposure to Oil, GER, and 2H. The circles constitute individual data, and the horizontal bars represent the medians of each distribution. Odor discrimination did not change over time after exposure to pheromone components.

For each bee responding at least to the highest sucrose concentration (30%), we quantified an individual sucrose-responsiveness score (SRS) as the number of sucrose concentrations that elicited a PER. Higher SRSs reflect a higher appetitive motivation. Preexposure to GER and 2H modified sucrose responsiveness (Kruskal–Wallis test, $H = 71.17$, d$f = 2$, $p < 0.001$, Fig. 3); specifically, GER-preexposed bees responded more to sucrose than controls, i.e., had higher SRSs (Dunn's post hoc test, $p = 0.0007$), while 2H preexposed bees responded less, i.e., had lower SRSs (Dunn's post hoc test, $p < 0.0001$), consistently with prior findings[30]. A population analysis confirmed that sucrose responsiveness increased with sucrose concentration (GLMM, sucrose concentration: $\chi^2 = 408.9$, d$f = 1$, $p < 0.0001$), and was higher in GER-preexposed bees and lower in 2H preexposed bees (treatment: $\chi^2 = 360.6$, d$f = 2$, $p < 0.0001$, Tukey's post hoc test, $p < 0.0001$ in both cases). SRSs are tightly related to learning success as higher SRSs correspond to better acquisition performances[31]. We thus studied if pheromone components modify this relationship. As expected, learning

success measured through individual acquisition scores (ACQS, see above) correlated with SRSs (Supplementary Fig. 3A, B): bees with low sucrose responsiveness (SRSs 1–2) had lower ACQSs, while bees with high sucrose responsiveness (SRSs 5–6) had higher ACQSs; intermediate SRSs (3–4) corresponded to inter-mediate ACQSs (Kruskal–Wallis test, $\chi^2 = 35.69$, d$f = 2$, $p < 0.001$). Yet, pheromone preexposure modified this relationship for bees with low (1–2) and high (5–6) SRSs (Kruskal–Wallis test, $SRS_{1-2}$: $p < 0.009$, $SRS_{3-4}$: $p = 0.17$, $SRS_{5-6}$: $p = 0.05$), showing that pheromone components modulate associative learning via their effect on appetitive responsiveness.

**Effect of pheromone components on aminergic signaling**. We next focused on neural signaling by octopamine (OA) and dopamine (DA) because these biogenic amines mediate appeti-tive[32] and aversive responsiveness[33], respectively, in the bee brain. We injected two doses of OA, DA, epinastine (OA-receptor antagonist), or flupentixol (DA-receptor antagonist) into the

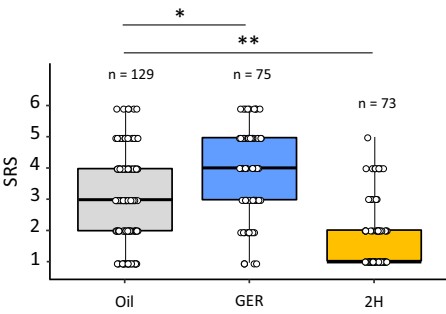

**Fig. 3 Pheromone components modulate sucrose responsiveness according to their valence.** Bees were preexposed to geraniol (GER, $n = 112$ independent bees), 2-heptanone (2H, $n = 156$ independent bees), or mineral oil ($n = 172$ independent bees). They were then stimulated with a series of six increasing concentrations of sucrose solution (0.1, 0.3, 1, 3, 10, and 30, w/w). For each bee that responded at least to the highest sucrose concentration (30%), we calculated an individual sucrose-responsiveness score (SRS) as the number of sucrose concentrations to which a bee responded. The figure shows the median, quartiles, and max and min (upper and lower whiskers) SRS values of bees preexposed to GER, 2H, or oil, and retained in the analyses. Individual bees are indicated by the dots. Preexposure to GER and to 2H induced a significant increase and decrease of SRS, respectively, with respect to bees exposed to mineral oil. (*) $p = 0.0007$; (**) $p < 0.0001$.

brain of bees that were preexposed to GER, 2H, or mineral oil prior to olfactory conditioning and retention tests (Fig. 4a). Phosphate-buffered saline (PBS) was injected into control bees. After preexposure to GER, which facilitates olfactory learning and memory, bees injected with epinastine (Fig. 4b) and with flupentixol (Fig. 4c) exhibited impaired learning and specific memory with respect to control bees (epinastine: acquisition: $\chi^2 = 12.46$, $df = 2$, $p = 0.002$; memory: $\chi^2 = 8.21$, $df = 2$, $p = 0.016$; flupentixol: acquisition: $\chi^2 = 7.74$, $df = 2$, $p = 0.02$; memory: $\chi^2 = 8.35$, $df = 2$, $p < 0.015$). GER-preexposed bees injected with OA or DA exhibited similar learning and specific-memory performances as control bees (see Supplementary Fig. 4A–C), consistent with a ceiling effect. These results indicate that the enhancing effect of GER preexposure on learning and memory was mediated by both octopaminergic and dopaminergic signaling.

After preexposure to 2H, which impairs olfactory learning and memory, injection of OA (Fig. 4d) rescued both learning ($\chi^2 = 18.20$, $df = 2$, $p < 0.0001$) and specific memory ($\chi^2 = 145.6$, $df = 2$, $p < 0.001$) with respect to control bees. Accordingly, epinastine did neither affect learning ($\chi^2 = 3.99$, $df = 2$, $p = 0.14$) nor memory ($\chi^2 = 0.032$, $df = 2$, $p = 0.98$) with respect to control bees (Supplementary Fig. 4D), consistent with a floor effect. Injection of DA (Supplementary Fig. 4E) did not affect olfactory learning in PBS-injected bees ($\chi^2 = 2.72$, $df = 2$, $p < 0.26$). Yet, DA enhanced specific memory (inj:test: $\chi^2 = 12.86$, $df = 2$, $p = 0.01$), similarly to OA. Injection of flupentixol (Fig. 4e) rescued partially the effects of 2H exposure. No effect on learning was detected ($\chi^2 = 0.11$, $df = 2$, $p = 0.94$, Fig. 4e), likely because 2H did not induce a clear effect in the PBS group. Yet, flupentixol improved significantly specific memory ($\chi^2 = 159.1$, $df = 2$, $p < 0.0001$), irrespective of the test and the dose used (Tukey's post hoc test, 2 mM vs. PBS: $p < 0.0001$; 0.2 µM vs. PBS: $p < 0.0001$). These results indicate that the depressing effect of 2H was in part mediated by dopaminergic signaling.

## Discussion
Our results reveal that pheromone components induce a persistent, valence-dependent modulation of learning and memory.

They do not affect odor processing but modulate appetitive motivation via aminergic circuits in the bee brain, thereby changing subsequent appetitive learning and memory formation. If bees in an appetitive-search mood are exposed to GER, which is the major component of the pheromone of the Nasonov gland used to attract individuals to sites of interest[18], they will learn more efficiently the features of that site. On the contrary, if they are exposed to 2H, which signals aversive events in various contexts[10,17,34], they may no longer be predisposed to learn about appetitive food cues[35], but rather to avoid or respond defensively to these events. The resulting scenario is adaptive as it improves foraging and orientation, and tunes responses toward relevant stimuli, even when the pheromone perceived is no longer present.

The effect of pheromone components on sucrose responsiveness[30] relied on aminergic modulation of appetitive motivation (see Supplementary Table 1 for a summary). GER acted on both octopaminergic and dopaminergic pathways. While the participation of OA in the enhancement of appetitive responses was predictable based on prior results[32,36,37], the finding that DA has a similar effect represents a novelty in the case of the honey bee. It suggests that appetitive signaling may also recruit the dopaminergic pathway, similarly to the case of the fruit fly, *Drosophila melanogaster*[38,39], where sucrose signaling occurs via a specific cluster of dopaminergic neurons. Evidence for a similar neural representation of sucrose has remained elusive in the honey bee until now. Yet, the strict separation between octopaminergic and dopaminergic signaling as mediating mechanisms of appetitive and aversive responsiveness in bees, respectively, needs both to be reconsidered and clarified. On the one hand, DA has been shown to impair appetitive memory consolidation in olfactory PER conditioning, while blockade of DA receptors enhances olfactory memory[40]. On the other hand, in experiments using a visual version of PER conditioning[41], DA-receptor blockade impaired appetitive visual learning and memory, while DA administration improved them. In our experiments, the impairment of learning and memory induced by 2H was counteracted by OA but not by epinastine. Inhibiting DA signaling via flupentixol left acquisition intact but improved memory retention, irrespective of the dose used, while the lower dose of DA also enhanced memory retention, thus showing that the effect of 2H on dopaminergic signaling requires further clarification.

Other molecular actors differing from biogenic amines may participate in the modulation of learning and memory induced by pheromones. For instance, isoamyl acetate (IAA), the main component of the sting alarm pheromone of honey bees, activates the equivalent of an opioid system, and decreases aversive responsiveness in a process that resembles stress-induced analgesia[42,43]. Exposure to both the sting alarm pheromone and to IAA alone impairs appetitive olfactory PER conditioning in a dose-dependent manner, but leaves sucrose responsiveness intact[35]. Learning impairment is mediated probably by allatostatins, in particular by the C-type allatostatin (ASTCC), which might be therefore responsible for the stress-induced analgesia[44]. If and how allatostatins interact with aminergic pathways to modulate learning and memory remains to be determined. Irrespective of the specific pathway involved, the result is consistent with our scenario: exposure to an aversive pheromone component decreases significantly appetitive learning.

Our findings differ from prior works suggesting that in newborn rabbits[5] and in adult rats[3,4], pheromone components can substitute unconditioned stimuli and mediate learning when paired with conditioned odorants or context. In our work, pheromonal components were neither present during learning, nor did we show that GER or 2H replace sucrose. We showed instead that preexposure to these substances changes appetitive motivation, and that this change affects subsequent appetitive

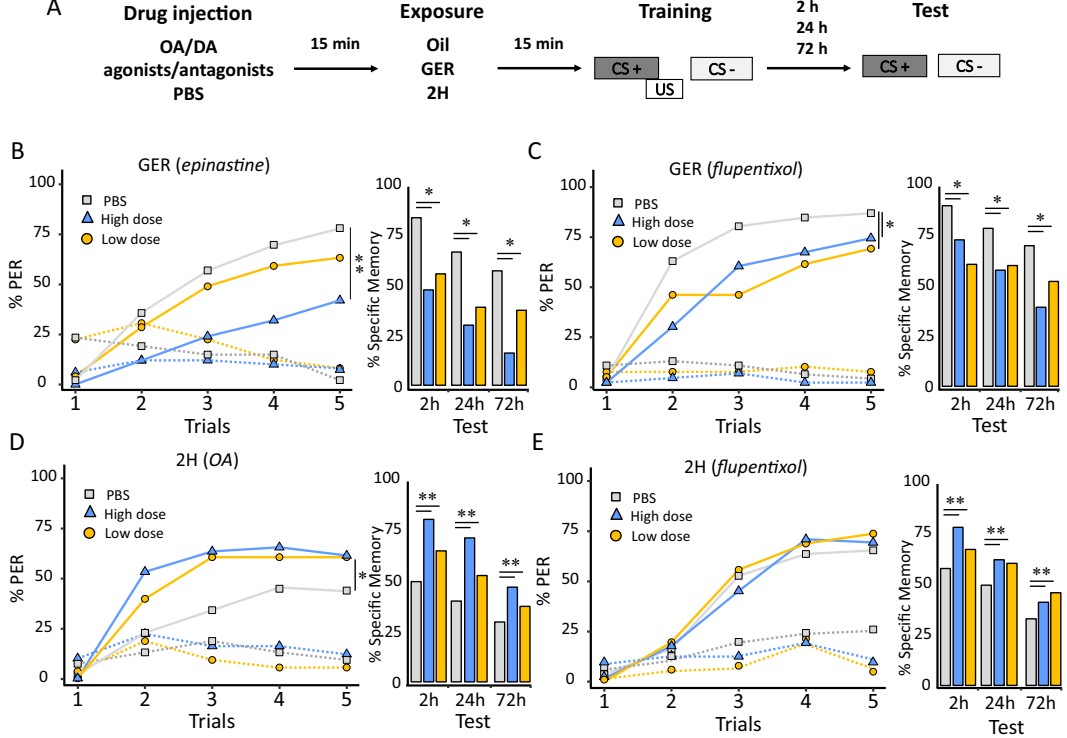

**Fig. 4 Effect of pheromone components on aminergic signaling in the bee brain. a–e** Pharmacological treatment with agonists/antagonists of the dopaminergic and octopaminergic system of honey bees exposed to either GER or 2H counteracted totally or partially the effects of pheromone components on learning and memory. **a** Experimental protocol used. **b** Proportion of conditioned responses (PER) to the rewarded (solid lines) and nonrewarded odors (dotted lines) during five CS+ and CS− trials, and proportion of bees with specific memory (i.e., the proportion of bees responding to the CS+ and not to the CS−) in retention tests performed 2, 24, or 72 h after conditioning (bar diagram) in the case of bees injected either with PBS ($n = 43$ independent bees, controls) or epinastine (OA-receptor antagonist) [0.4 µM ($n = 42$ independent bees), 4 mM ($n = 45$ independent bees)] and preexposed to geraniol (GER). (*) $p < 0.05$; (**) $p \leq 0.001$. **c** Same as in **b** but for bees injected either with PBS ($n = 45$ independent bees) or flupentixol (DA-receptor antagonist) [0.2 µM ($n = 40$ independent bees), 2 mM ($n = 43$ independent bees)] and exposed to geraniol (GER). Both the OA and the DA-receptor antagonist counteracted the enhancing effect of GER on learning and memory formation. (*) $p < 0.05$. **d** Same as in **b** but for bees injected either with PBS ($n = 53$ independent bees) or with octopamine [20 µM ($n = 53$ independent bees) and 2 mM ($n = 49$ independent bees)] and preexposed to 2-heptanone (2H). OA treatment counteracted the decrement of learning and memory induced by 2H. (*) $p < 0.05$; (**) $p \leq 0.001$. **e** Same as in **b** but for bees injected either with PBS ($n = 55$ independent bees) or with flupentixol (DA-receptor antagonist) [0.2 µM ($n = 61$ independent bees) and 2 mM ($n = 62$ independent bees)] and exposed to 2-heptanone (2H). The DA-receptor antagonist had no effect on learning but rescued memory retention. (**) $p \leq 0.001$.

learning in the absence of the pheromonal signal. If our pheromone components would simply replace reinforcement, their preexposure should retard olfactory conditioning according to the known US-preexposure effect[45]. Clearly, this was not observed upon GER preexposure.

In the context of queen dominance, homovanillyl alcohol, a main component of the queen mandibular pheromone, impaired aversive but not appetitive learning of young bees[46], an effect that was considered specific of the queen–nurse interaction[47]. Also, isoamyl acetate, the main component of the sting alarm pheromone, impaired appetitive learning, which was considered specific of the sting alarm pheromone[35]. Our findings show that these cases were not specific but reflect a generalized effect on learning that extends across ages and behavioral contexts.

These examples highlight the common use of single, major pheromone components to study the effects of pheromones that integrate more compounds in their blends. We used GER, which is the major component of the attractive pheromone of the Nasonov gland, and has an attractive effect per se[10]. GER is also part of some floral scents, as is the case of many chemicals that are shared between plants and pheromones[48]. Yet, in contrast to the minute quantities of GER available in plant scents (expressed in ppm)[49], the amount of GER used in the present work constituted a massive signal corresponding to several bees recruiting

for an appetitive event. In the case of 2H, the amount used corresponded to 1–3 mandibular glands of foragers[50]. Although the value of 2H as alarm signal in collective defensive responses is debated[10,51], its use to signal negative events has been verified[10]. Thus, the use of single pheromone components differing in valence is a valid strategy to estimate how pheromones affect a receiver's behavior according to their valence.

Pheromones evolved to act as chemical signals, which are effective because the receiver's response evolved accordingly. Our results indicate that besides the primary effect of conveying an intraspecific message, pheromones have a secondary effect in the receiver, which may have evolved with the primary one. This effect is the adaptive modulation of learning and memory in a way consistent with the valence of the pheromone perceived. We thus suggest that the definition of pheromone should incorporate the capacity of these substances to act as behavioral modulators of both motivational and cognitive processes.

## Methods

**Experimental animals and subject preparation.** Honey bee workers (*Apis mellifera*) were reared in outdoor hives at the experimental apiary of the CRCA situated in the campus of the University Paul Sabatier. In all cases, honey bee foragers (2–3 weeks old) were used. Foragers were collected each day at a feeder and immediately brought to the laboratory where they were cold-anesthetized for 5 min and gently restrained within a metal holder. Adhesive tape was used to block

the bees. Their heads were fastened to the holder with a drop of low-temperature melting wax so that only antennae and mouthparts could be moved[22]. Harnessed bees were either fed with 5 µl of sucrose solution (50% w/w) or fed to satiation depending on the experiment (see below). After feeding, bees were kept resting in a dark and humid place (ca. 60%) at $25 \pm 1$ °C until the start of the experiments. Through this procedure, we aimed at equalizing the hunger level across individuals, and at keeping bees with sufficient appetitive motivation and endurance[22].

**Pheromone-component exposure**. All pheromone substances were diluted to 24% in mineral oil (6-µl pheromone + 19-µl mineral oil), following a standard procedure[35]. Hence, the dissolvent, mineral oil, served as a control when presented alone. Restrained bees were confined within a 35-ml glass vial containing a filter paper ($1 \times 5$ cm) soaked with either 25 µl of mineral oil or the diluted pheromone component (Geraniol, GER, or 2-Heptanone, 2H). Doses of pheromone components were based on previous studies showing modulation of stimulus responsiveness[30,35,52], and correspond to the natural situation of several bees signaling a target. The exposure time (15 min) was shown to be sufficient to trigger different physiological and behavioral changes[30,35,52] so that we hypothesized that it could also translate into changes in learning and memory. Pheromone exposure was followed by a rest period that lasted 15 min. All chemicals were purchased from Sigma-Aldrich (France).

**Sucrose-responsiveness assay**. Harnessed bees collected at 5:00 p.m., fed to satiation, and kept resting overnight, were used the day after to quantify their sucrose responsiveness by recording PER in response to increasing concentrations of sucrose, following a standard protocol[28,29]. To this end, early in the morning of the testing day, bees received an additional 5 µl of sucrose (50% w/w) and were kept resting for 1 h. After resting and before mineral oil or pheromone-component exposure, bees were allowed to drink water ad libitum in order to ensure that they would respond only to the sucrose contained in the solutions assayed. Fifteen minutes after the end of exposure, both antennae of each bee were stimulated by means of a toothpick with six sucrose solutions of increasing concentration: 0.1, 0.3, 1, 3, 10, and 30% (w/w). Sucrose of analytical grade (Sigma-Aldrich, France) diluted in deionized water (Milli-Q system, Millipore, Bedford, USA) was used to prepare the solutions. Antennal stimulations with deionized water were interspersed between successive sucrose stimulations to avoid sucrose sensitization. The interstimulus interval was 2 min. Bees that did not respond to any sucrose concentration, that responded to water, or that exhibited inconsistent responses to sucrose (i.e., responding to lower but not to higher sucrose concentrations) were discarded[30]. An individual sucrose-responsiveness score (SRS) was obtained for each bee based on the number of responses (PER) to the six sucrose concentrations assayed. A bee with a SRS of 1 only responded to the highest concentration (30%) but not to the lower ones. A bee with a SRS of 6 responded to all six sucrose concentrations, including the most diluted ones.

Within each group, only a small percentage of bees (3.8–9.8%) were discarded as they responded to water or exhibited inconsistent responses to successive sucrose concentrations. The proportion of discarded bees did not differ between groups ($\chi^2 < 1.56$, d$f = 1$, $p > 0.20$ for all pairwise comparisons). Among the remaining bees, some failed to respond to any sucrose concentration, and they did so more often following preexposure to 2H (34%) than to mineral oil (3.5%) (2H vs. oil: $\chi^2 = 51.54$, d$f = 1$, $p < 0.0001$). GER had no effect (4.5%) on the absence of responsiveness (GER vs. oil: $\chi^2 = 0.17$, d$f = 1$, $p = 0.67$). The bees used for assessing the effect of pheromone components on sucrose responsiveness were afterward used for determining the effect of the same components on olfactory learning and retention.

**Olfactory conditioning and retention tests**. Bees tested for sucrose responsiveness were kept in a dark and humid place for one additional hour, and then exposed for a second time to the same pheromone component (or to mineral oil) during 15 min as described above. Two hours elapsed between the two exposures. Previous results showed that the modulatory effects of pheromone components on sucrose responsiveness were consistent and nonadditive over successive exposures[30]. Fifteen minutes after the end of the exposure, bees were subjected to olfactory PER conditioning in the form of a differential conditioning[14,22]. Bees were trained to discriminate a rewarding odorant (CS+) from a nonrewarding odorant (CS−) during ten trials (5 CS+ trials and 5 CS− trials) presented in a pseudorandom sequence so that the same stimulus (CS+ or CS−) was never presented more than twice consecutively. A 12-min intertrial interval was used. Sucrose solution (30%, w/w) delivered by a toothpick to the bees' antennae and proboscis was used as appetitive US. Limonene and eugenol (Sigma-Aldrich, France) were used as conditioned odorants (CSs). Both odors were used either as CS+ or CS− in a counterbalanced design. Four microliters of the pure odors were added each to a filter paper ($0.4 \times 4$ cm) placed into a 1-ml syringe connected with a computer-controlled odor-stimulation device, which allowed an efficient temporal control of the odor stimulation. Each acquisition trial lasted 30 s. It consisted of a 13-s familiarization phase with the automated odor releaser and the experimental context, a 6-s forward-paired presentation of the CS and the US in the case of CS+ trials (odorant and sucrose presentations lasted 4 s and 3 s, respectively,

with a 1-s overlap), and a 11-s resting phase in the setup. CS− trials followed the same sequence, but no sucrose was delivered upon odorant presentation. Memory tests were performed 2, 24, and 72 h after the end of the conditioning experiments. The tests consisted in CS+ and CS− presentations as in the training phase, but in the absence of US. Each odor was presented during 4 s with the same timing used for the conditioning trials. Odor presentations were separated by an intertrial interval of 12 min. The order of presentation of the CS+ and CS− was randomized between bees. Bees were fed to satiation and kept resting in a dark and humid place (ca. 60%) at $25 \pm 1$ °C at least 30 min after the end of the 2-h retention test, and once each other day. Although testing the same bees more than once can induce memory extinction, we adopted this procedure in the experiment in which exposure was performed before conditioning. We assumed that this phenomenon, if any, might affect both the experimental and the control groups, which were run in parallel. Bees that did not respond to the CS− or to the CS+ during the tests were stimulated with sucrose to check the integrity of the unconditioned response. Bees not responding to sucrose at the end of the retention tests were discarded[22].

To study specifically the effect of pheromone components on memory retrieval, a process different from memory formation, we exposed bees to pheromone/ mineral oil *after* conditioning and before the memory tests performed 2, 24, and 72 h after conditioning. Retention was assessed in groups of bees that had reached equal learning levels prior to exposure and established an associative memory. Independent groups of bees were used for each memory test so that each bee was assessed only once. An identical experiment was repeated in different groups of bees exposed twice to either pheromone component (GER or 2H) or mineral oil as control to ensure that the number of exposures prior to the memory test did not affect memory retrieval (Supplementary Fig. 5A, B).

**Calcium imaging of olfactory coding in the ALs**. Animal preparation and in vivo calcium imaging of projection-neuron (PN) activity at the level of the ALs upon stimulation with limonene and eugenol were performed as described previously[53]. Foragers were collected at 5:00 p.m. and harnessed in Plexiglas stages designed for calcium imaging experiments. After feeding them with 10 µl of sucrose solution (50% w/w), a small window was opened in the head cuticle. Glands and trachea covering the injection site were gently removed. A fluorescent calcium-sensitive dye (Fura-2 conjugated with dextran, Thermo-Fisher Scientific) was injected into the medial and lateral antenno-protocerebral tracts of PNs. To this end, the dye, crystallized at the tip of a pulled borosilicate glass needle, was manually inserted between the MB calyces, below the alpha lobe[53]. The left and right ALs were prepared alternately to avoid potential biases due to lateralization. The cuticle was replaced on top of the head and sealed with n-eicosane to avoid brain desiccation. Bees were fed to satiation and left in a dark humid chamber overnight. On the following morning, after the dye had diffused retrogradely toward the PN dendrites, bees received 5 µl of sucrose solution (50% w/w), the head capsule was reopened, and the glands and trachea were removed to expose the AL. The brain was covered with a transparent two-component silicon (Kwik-Sil, WPI). Imaging of the AL responses was performed under a two-photon microscope (Ultima IV, Bruker) equipped with a 20× water-immersion objective (NA 1.0, Olympus)[53]. Fluorescence intensity was recorded along a custom scanline of interest, recording all glomeruli in a given focal plane. Bees were exposed to three alternating stimulations of limonene and eugenol for 4 s each, with an interstimulus interval of 13 s (corresponding to the familiarization time used in the conditioning experiment). Fifteen minutes after this first imaging session, bees were exposed to mineral oil or to one of the two pheromones (GER, 2H) for 15 min. Two additional imaging sessions were performed 15 min and 2 h after pheromone/mineral-oil exposures. The latter period corresponds to the end of conditioning in trained animals. The potential impact of pheromone exposure on odorant coding and differentiation was evaluated by comparing the overall activation signal averaged over all observed glomeruli, and the distinguishability of the odor-coding patterns quantified in terms of the Euclidean distance. Data analyses were performed with MATLAB (R2018, MathWorks). Glomerular response signals were deduced from relative changes in fluorescence, with respect to a background signal: $-\Delta F(t)/F$. The background fluorescence $F$ was obtained in a dynamic way by applying a moving average filter with a long time constant (7 s) to the fluorescence time series. Ten glomeruli were identified for each bee. Glomerular responses were averaged over the first 600 ms of olfactory stimulation and over the three repeated stimulations for each odorant.

The ED between the response patterns of two odors $x$ and $y$ was calculated by the following formula:

$$\mathrm{ED}_{x,y} = \sqrt{\sum_{i=1}^{n} (x_i - y_i)^2},$$

where $x_i$ and $y_i$ are the average responses of the single glomerulus $i$ to $x$ and $y$, which are summed over all $n$ glomeruli observed in a single bee.

**Neuropharmacological experiments**. Bees used in the neuropharmacological experiments were collected each day at 9.00 a.m. (see above in the "Subject Preparation" section for detailed information) and subjected to drug administration, pheromone exposure, and olfactory conditioning of PER during the same day.

Octopamine (OA) and dopamine (DA) and their respective receptor antagonists epinastine hydrochloride (epinastine)[54] and cis-(Z)-flupentixol dihydrochloride (flupentixol)[55] (Sigma-Aldrich, France) were injected into the bee brain via the ocellar tract[20]. Injections were performed after gentle removal of the median ocellus with a scalpel. This procedure allows the drug to reach and dissolve uniformly into the protocerebrum within few minutes. Injections were done using a WPI 26 gauge needle of a NanoFil 10-μl syringe controlled by a micromanipulator (M3301R, WPI) under a binocular stereomicroscope (Leica)[20]. To test for dose–response effects, each substance was used at a lower and a higher dose (OA: 20 μM and 2 mM; DA: 20 μM and 2 mM; epinastine: 0.4 μM and 4 μM; flupentixol: 0.2 μM and 2 mM). Drugs were dissolved in PBS, which was injected (200 nl) alone into control bees. For each exposure treatment (GER, 2H, or mineral oil) and each drug injected (agonist or antagonist), low-dose, high-dose, and PBS-injected groups were run in parallel. Bees losing hemolymph after surgery or not showing drug penetration after a couple of minutes were discarded. Each day, before and after injections, syringes and needles were washed using ethanol and distilled water. Injections were performed 15 min before pheromone exposure (see above) and 30 min before olfactory PER conditioning. Previous studies showed that 30 min are required for these biogenic amines and their antagonists to be effective[20,56]. Protocols for exposure, conditioning, and memory-retention tests were the same as those previously described.

**Statistics and reproducibility**. Sample sizes are provided in each figure caption for the different types of experiments. The behavioral readout used throughout was the PER (1 or 0) to the US (sucrose-responsiveness experiment) or to the CS− and CS + (learning and memory experiments). The percentage of bees responding to the US or the CSs was calculated and represented as a population response. Individual scores based on the number of PER to a given number of stimulations (either US or CS) were also computed. A SRS was calculated for each bee as the number of sucrose concentrations eliciting PER. Differences between the SRSs of different groups of bees were analyzed using a Kruskal–Wallis test followed by Dunn's pairwise test for multiple comparisons. Repeated-measure ANOVA was used to analyze behavioral performances in sucrose-responsiveness assays and in olfactory conditioning of bees exposed to pheromone components and/or treated with drugs. Independent models were used for the acquisition phase and the memory test for bees exposed either before or after conditioning. Individual conditioned responses were examined using GLMMs with a binomial error structure—logit-link function—*glmer* function of R package *lme4*[57]. When necessary, models were optimized with the iterative algorithm BOBYQA[58]. In the sucrose-responsiveness model, "bee response" was entered as a dependent variable, "treatment" (GER, 2H, or mineral oil) as a fixed factor, and "sucrose concentrations" as covariates. In the learning models, "bee response" was entered as a dependent variable, "treatment" (GER, 2H, or mineral-oil exposure) and "CS" as fixed factors, and "conditioning trial" as covariate. In the memory-retention models for bees exposed before conditioning, "bee response" was entered as a dependent variable, and "treatment" and "testing time" (2, 24, or 72 h) as fixed factors. In all cases, "individual identity" (IDs) was considered as a random factor to allow for repeated-measurement analysis.

An acquisition score (ACQS) was calculated for each bee as the sum of responses to the five CS+ presentations during conditioning. We analyzed if pheromone-component exposure changed the ACQS of bees pertaining to different SRS categories. To this end, bees were grouped into three groups based on their SRSs (SRS$_{1-2}$, SRS$_{3-4}$, or SRS$_{5-6}$), and their ACQSs were compared using a Kruskal–Wallis test followed by Dunn's pairwise test for multiple comparisons. Memory retrieval at 2, 24, and 72 h of independent groups of bees exposed after conditioning was analyzed using a Binomial Logistic Regression test. In the model, "bee response" was entered as a dependent variable and "treatment" (GER, 2H, or mineral-oil exposure) as a fixed factor. For the neuropharmacological experiments, independent models were carried out either for the acquisition phase and the memory test or for each exposure treatment. These GLM models were designed as described above with the exception that the fixed factor "treatment" (exposure) was replaced by the fixed factor "pharmacological treatment". In all the GLM models, "individual identity" (IDs) was considered as a random factor to allow for repeated-measurement analysis. In many analyses, several models were run and compared to identify significant interactions between fixed factors and/or covariates, and the significant model with the highest explanatory power (i.e., the lowest Akaike information criterion (AIC) value) was retained. When AIC values were very similar, the most significant model was retained. Interactions, wherever significant, are indicated in the text. Tukey's post hoc tests were used to detect differences between the different groups (*glht* function from R package *multcomp*[59]). All statistical analyses were performed with R 3.4.2[60].

**Reporting summary**. Further information on research design is available in the Nature Research Reporting Summary linked to this article.

## Data availability
The datasets generated during this study are available at figshare.com with the following accession ID: https://doi.org/10.6084/m9.figshare.12029526.v1[61].

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

## Acknowledgements

We thank Alex Kacelnik for useful discussions on the role of pheromones. This work was supported by the French National Research Agency (Project PHEROMOD, ANR-14-CE18-0003), the Institut Universitaire de France, and the IDEX Program "Excellence Chairs" of the Université Fédérale Toulouse Midi-Pyrénées. A.C. and A.H. acknowledge funding by the Autonomous Province of Bolzano (Project B26J16000310003). The funders had no role in study design, data collection and interpretation, or the decision to submit the work for publication.

## Author contributions

D.B. performed the behavioral and the pharmacological experiments, and analyzed the corresponding data. A.C. performed the calcium imaging experiments and analyzed the data. M.G. and P.d.'E. supervised the behavioral and the pharmacological experiments. A.H. supervised the calcium imaging experiments. All authors (D.B., A.C., J.-M.D., A.H., P.d.'E., and M.G.) helped design the experiments. D.B. and M.G. wrote the paper. All authors participated in the editing of the paper. M.G. and P.d.'E. obtained the principal funding for the research.

## Competing interests

The authors declare no competing interests.
