## [Peer Review File · Communications Biology]

Reviewers' comments:

Reviewer #1 (Remarks to the Author):

I enjoyed reading this manuscript and thought that the experiment was an interesting one. However, I feel that the introduction would benefit from setting it in a broader setting, and more discussing the broader relevance of the questions being asked. I also think the introduction needs to clearly explain how this work builds off previous work and what exactly the hypotheses are being tested – and what the predictions are. The difference between the two experiments reported on lines 103 and 113 are interesting, but this isn't well pre-empted in the introduction, so the significance is not immediately clear.

I also think the manuscript would benefit from an overview diagram of the experimental methods (e.g. when the bees are exposed to the pheromone in relation to conditioning, and testing etc.)

My only other major comment relates to the statistical analysis:

Line 400 – it seems like by analysing the sucrose responsiveness data this way you are losing information about the different concentrations of sucrose? Why not do a binomial mixed model where the dependent is whether the responded or not (0 or 1) and then include the independent variables 'sucrose concentration', 'treatment' and the random factor 'bee'. By including all the factors in a single model you also would not have to then control for multiple statistical tests.

Minor Comments:

Line 82- make it clear why bees were pre-exposed to mineral oil.

Line 104- I think it would make it clearer to state here when the bees were exposed to the pheromone in relation to conditioning (i.e. before)

Line 108 – but there wasn't a significant interaction in this case?

Line 111-112 – this is a helpful summary sentence of your results and I think should go at the beginning of this section before you report the stats.

Line 263 – why was mineral oil used as a control? It also contains volatiles that bees can respond to

Line 317 – in the test phase, how long were the bees given to respond to the odors?

Line 410 – statistical package(s) need to be cited and included in reference list

Line 416 – “we analyzed if pheromone exposure modifies this correlation in a significant way” – can you make it clearer why you did this.

Figure 1- I think the graph would be clearer to interpret if you used different colors (it took me a second to work out the difference between 'sky blue' and 'light blue'!)

Reviewer #2 (Remarks to the Author):

How motivational states affect behaviour has been a topic of main interest for psychologists and ethologists for decades and lots of evidence indicate that motivation can profoundly influence learning in a variety of species from invertebrates to humans. This study by Baracchi et al. investigates the question of how motivational systems are organised and how they impact neural circuits involved in learning, which is a major but much less explored area in neurobiology.

The study utilises the previous knowledge that different pheromones are perceived by honey bees as either attractants (Geraniol) or deterrents (2-heptanone) and investigates how the motivational states induced by these chemicals can influence olfactory learning and memory retention. The results show that modulation of motivational states by pheromones affect learning and memory formation without influencing the CS response itself when applied after the learning. The authors relate the enhanced/diminished learning to the altered motivational state caused by the pheromones leading to a change in responsiveness sucrose (used as the US in the associative classical conditioning paradigm). This is an important discovery since it proves that motivational states influence cognitive processes rather than simply affecting the responsiveness of the feeding circuitry. These results were also confirmed by using in vivo two-photon calcium imaging to monitor neuronal activity in the antennal lobes and to test the effect of pheromones on aminergic circuits mediating appetitive responsiveness. Overall the work significantly contributes to the understanding of mechanisms underlying the influence of motivational state on learning and memory retention.

The experiments are carefully designed and the results are built on solid data so I have no major concerns with the paper.

Minor comments

1. The results shown in Fig 1. illustrate that GER-exposed bees learned better while H2 exposed ones learned worse than Oil exposed ones during 5 training trials. The results were also compared to unrewarded exposure to the different pheromones/oil. In a previous paper the authors demonstrated that repeated exposure to sucrose (the US in the present experiments) causes habituation and this is also influenced by exposure to the pheromones. A comment on how this might also be a component that affects the classical conditioning (multiple exposures to sucrose during the training trials) used in the current experiments would be an interesting addition.
2. At 72h the decrease in memory retention is still significant after 2H exposure while it becomes non-significant when animals experienced GER exposure. Could this be because the negative effect of H2 is stronger than the positive effect of GER? (In fig 3. it is indicated that significance of the 2H effect is stronger.)
3. The results illustrated in Fig. 3. show that individual sucrose responsiveness scores (SRS) were differentially affected by exposure to GER or H2. However it is not clear whether the numbers on the Y-axes correspond only to the number of responses or do they also correlate to the concentration strength. (I.e. 1 represents the highest concentration and the number of responses to that or even if the response was given to the lowest concentration the individual score was still 1.)
4. In rows 218-220 the authors state that: "While the participation of OA in the enhancement of appetitive responses was predictable based on prior results the finding that DA has a similar effect represents a novelty." While it is true that there are indications from other works that although OA and DA are the main regulators of appetitive and aversive learning, respectively, in insects they are not restricted to only one kind of learning but can also modulate the opposite one (e.g. Klappenbach et al 2013). This could also be part of the discussion.
5. In the introduction and discussion the findings could be more related to the wider context of how motivational states can affect learning and memory not just in invertebrates but other species including humans.
6. The expression "in a durable way" is used in row 52 and 72 but it is not entirely clear what the authors mean by this.

7. In row 82 "stating" should be starting.
8. In row 103 instead of "memory formation" it may be more appropriate to use "memory retention".
9. in Supplementary Table 1 the signs should be arranged more clearly showing the corresponding signs in all the rubrics.

Reviewer #3 (Remarks to the Author):

The authors have asked a fascinating question, if pheromones, chemical messengers within an species, can influence cognitive processes. I think that the results are clearly presented and the experiments are appropriately designed and conducted. I believe that the effects are real, but I have some concerns about calling 2-heptanone and geraniol pheromones.

2H is a component of mandibular alarm pheromone, but there is lack of good evidence that it has an alarm function. For example, the Shearer and Boch (1965) paper showed repellent effects (bees avoided alighting at the nest entrance) but did not report the amount of the compound that they used. In Boch and Shearer (1971), which is a better reference because it provides more quantitative details, they explain that they provided $14 \pm 2 \mu\text{l}$ of the test substance (they tested multiple ones, including 2-H) onto a small cork, as described in their 1965 paper. However, this is not a realistic concentration, and the aversion could be due to simply the very high amount that this represents. Papachristoforou et al. (2012) reported that each honey bee had an average of $0.0386 \mu\text{l}$ of 2-H. If this true, then Boch and Shearer (1971) tested 362 bee equivalents of 2-H. If we look at Papachristoforou et al. (2012), the function of 2-H is not alarm but rather a toxin injected into the bodies of hive parasites. In this case, I suppose it could serve as a warning message to nestmates who smelled it, but Papachristoforou et al. (2012) did not demonstrate this.

There is much stronger evidence for isopentyl acetate (IPA) as an alarm pheromone component, and I imagine that the authors would achieve a similar effect with this compound. However, I would still caution them about the use of the term "pheromone", which typically refers to a highly specific blend of semiochemicals for which the exact proportions are usually highly relevant.

Similarly, although GER is a component of the Nasanov gland, which produces a blend of compounds that are a true pheromone, it is unclear if GER is, by itself, a true pheromone. GER is a common volatile in the headspaces of flowering plants that are visited by bees for nectar and pollen. Given that the authors have demonstrated that the effect of GER is to enhance appetitive motivation, there seems no need to describe GER as a pheromone. Rather, it a common odor that is associated with food.

I think that the manuscript is quite interesting and still worthy of publication, but I suggest that the authors reduce their claims about these being pheromones. At best, they are pheromone components and for GER it not clear if the effect is due to GER being an odor associated with food or a component of Nasanov pheromone (but not the entire pheromone itself). I understand that making the claim that these are pheromones increases the potential impact of the study, but the fact that certain odor compounds can alter appetitive motivation in bees is a simpler claim that more closely follows what the authors have found.

A more minor question is why the pre-exposure interval of 15 min chosen? Is there a biological basis for this?

Responses to Reviewers

Reviewer #1

I enjoyed reading this manuscript and thought that the experiment was an interesting one. However, I feel that the introduction would benefit from setting it in a broader setting, and more discussing the broader relevance of the questions being asked. I also think the introduction needs to clearly explain how this work builds off previous work and what exactly the hypotheses are being tested – and what the predictions are. The difference between the two experiments reported on lines 103 and 113 are interesting, but this isn't well pre-empted in the introduction, so the significance is not immediately clear.

- We thank the Reviewer for her/his overall positive comments on our manuscript. Concerning the request of setting our Introduction in a broader setting, we would like to call her/his attention to the length constraints of the journal, which impose concision and sometimes the impossibility of expanding arguments. We have tried, nevertheless, to set the Introduction in a broader context while respecting as much as possible the length constraints of the journal. We have explained how our work builds off previous work (lines 53-58), and have explained the difference between the concepts of memory formation and memory retrieval to understand better the experiments mentioned by the Reviewer (lines 73-76).

I also think the manuscript would benefit from an overview diagram of the experimental methods (e.g. when the bees are exposed to the pheromone in relation to conditioning, and testing etc.).

- We thank the reviewer for this useful suggestion. We have added such a diagram on top of each figure displaying behavioral results (see Figs 1, 3 and 4, and Figs S1-S5). We hope that this clarifies our methodology and results.

My only other major comment relates to the statistical analysis:

Line 400 – it seems like by analysing the sucrose responsiveness data this way you are losing information about the different concentrations of sucrose? Why not do a binomial mixed model where the dependent is whether the responded or not (0 or 1) and then include the independent variables 'sucrose concentration', 'treatment' and the random factor 'bee'. By including all the factors in a single model you also would not have to then control for multiple statistical tests.

- We thank the Reviewer for this suggestion. We have added the suggested GLMM analysis, which confirmed that the treatment affected sucrose responsiveness (treatment: $\chi^2 = 360.6$, $df = 2$, $p < 0.0001$) in the way previously reported: geraniol enhanced it (Tukey post hoc, $p < 0.0001$), while 2H decreased it ($p < 0.0001$). The analysis also showed that overall responsiveness increased with increasing sucrose concentrations (GLMM, Sucrose concentration: $\chi^2 = 408.9$, $df = 1$, $p < 0.0001$). We have included this analysis (see lines 182-185) and details about the model in the methods section (lines 453-455) of the new version.

Minor Comments:

Line 82- make it clear why bees were pre-exposed to mineral oil.

- We have specified that mineral oil is a standard control stimulus in studies on insect olfaction (lines 95-96, 300-301).

Line 104- I think it would make it clearer to state here when the bees were exposed to the pheromone in relation to conditioning (i.e. before)

- We have clarified this point (see lines 116-118).

Line 108 – but there wasn't a significant interaction in this case?

- Indeed, in the model the interaction was not significant (GLMM, treatment*testing time: $\chi^2 = 6.69$, $df = 4$, $p = 0.153$). Accordingly, the post hoc tests run were on the main factor treatment (GER, 2H and mineral-oil exposure). We have specified this in the new version (see line 123).

Line 111-112 – this is a helpful summary sentence of your results and I think should go at the beginning of this section before you report the stats.

- We thank the Reviewer for this suggestion. We have followed her/his advice and moved the sentences before the statistical part (now lines 120-121).

Line 263 – why was mineral oil used as a control? It also contains volatiles that bees can respond to

- Mineral oil is commonly used as a standard control in experiments on insect olfaction¹⁻³. It is a solvent commonly used to equalize the vapor pressure of odorants to be tested. In our case, all pheromone components were diluted to 24% in mineral oil. Hence, mineral oil alone was used as a control for these dilutions. We are not aware of experiments reporting that honey bees respond to volatiles contained in mineral oil using mineral oil as the stimulation source. We have provided these explanations in lines 300-301.

Line 317 – in the test phase, how long were the bees given to respond to the odors?

- Odor presentation during the tests followed the same timing as in the conditioning trials, i.e. each odor was presented during 4 s. We have specified this in lines 360-361.

Line 410 – statistical package(s) need to be cited and included in reference list

- We have cited the statistical packages and included them in the reference list.

Line 416 – “we analyzed if pheromone exposure modifies this correlation in a significant way” – can you make it clearer why you did this.

- We thank the Reviewer for this question. The direct correlation between sucrose responsiveness scores (SRS) and learning scores (ACQS) has been demonstrated multiple times by the seminal work of Dr. Ricarda Scheiner. Briefly, bees with low SRS exhibit poor ACQS while bees with high SRS exhibit high ACQS. We analyzed if pheromone exposure modified the relation between SRS and ACQS in a significant way. In other words, we determined if pheromone exposure changed the ACQS of bees pertaining to different SRS categories, as an additional proof of their capacity to modify cognitive processes. Indeed, GER exposure enhanced ACQS in low SRS bees and tended to increase even more ACQS of high SRS bees. On the contrary, 2H decreased ACQS with respect to control bees. This constitutes a clear demonstration of the capacity of pheromones to modulate responsiveness and cognitive processes in bees. We have specified this in the revised version (see lines 186-188).

Figure 1- I think the graph would be clearer to interpret if you used different colors (it took me a second to work out the difference between 'sky blue' and 'light blue'!)

- We agree with the Reviewer and thank her/him for the remark. We have modified the colors of the figures to avoid this problem.

Reviewer #2 (Remarks to the Author):

How motivational states affect behaviour has been a topic of main interest for psychologists and ethologists for decades and lots of evidence indicate that motivation can profoundly influence learning in a variety of species from invertebrates to humans. This study by Baracchi et al. investigates the question of how motivational systems are organised and how they impact neural circuits involved in learning, which is a major but much less explored area in neurobiology. The study utilises the previous knowledge that different pheromones are perceived by honey bees as either attractants (Geraniol) or deterrents (2-heptanone) and investigates how the motivational states induced by these chemicals can influence olfactory learning and memory retention. The results show that modulation of motivational states by pheromones affect learning and memory formation without influencing the CS response itself when applied after the learning. The authors relate the enhanced/diminished learning to the altered motivational state caused by the pheromones leading to a change in sucrose responsiveness (used as the US in the associative classical conditioning paradigm). This is an important discovery since it proves that motivational states influence cognitive processes rather than simply affecting the responsiveness of the feeding circuitry. These results were also confirmed by using in vivo two-photon calcium imaging to monitor neuronal activity in the antennal lobes and to test the effect of pheromones on aminergic circuits mediating appetitive responsiveness.

Overall the work significantly contributes to the understanding of mechanisms underlying the influence of motivational state on learning and memory retention.

The experiments are carefully designed and the results are built on solid data so I have no major concerns with the paper.

- We thank the Reviewer for her/his overall positive comments on our manuscript.

Minor comments

1. The results shown in Fig 1. illustrate that GER-exposed bees learned better while H2 exposed ones learned worse than Oil exposed ones during 5 training trials. The results were also compared to unrewarded exposure to the different pheromones/oil. In a previous paper the authors demonstrated that repeated exposure to sucrose (the US in the present experiments) causes habituation and this is also influenced by exposure to the pheromones. A comment on how this might also be a component that affects the classical conditioning (multiple exposures to sucrose during the training trials) used in the current experiments would be an interesting addition.

- We thank the Reviewer for this comment. Following her/his advice, at the end of the behavioral experiments, we have specified that given that olfactory PER conditioning is a case of Pavlovian learning involving an olfactory CS and sucrose solution as the US, the modulatory effect of pheromones may have been exerted at the level of one or both components of the association learned (lines 138-140).

2. At 72h the decrease in memory retention is still significant after 2H exposure while it becomes non-significant when animals experienced GER exposure. Could this be because the negative effect of H2 is stronger than the positive effect of GER? (In fig 3. it is indicated that significance of the 2H effect is stronger.)

- In the case of SRS (Fig. 3), 2H had a stronger effect than geraniol. However, in olfactory conditioning, in particular in the fifth conditioning trial (Fig. 1A), geraniol had a stronger effect than 2H. Thus, we think that it is difficult to be conclusive on this point.

3. The results illustrated in Fig. 3. show that individual sucrose responsiveness scores (SRS) were differentially affected by exposure to GER or H2. However it is not clear whether the numbers on the Y-axes correspond only to the number of responses or do they also correlate to the concentration strength. (I.e. 1 represents the highest concentration and the number of responses to that or even if the response was given to the lowest concentration the individual score was still 1.)

- We thank the Reviewer for this question. The numbers (sucrose responsiveness scores, SRS) refer to the number of responses produced upon stimulation with a series of increasing sucrose concentrations (see definition in lines 177-178). This is the standard definition adopted in the field of studies on sucrose responsiveness in insects, including bees⁴⁻¹⁰, flies¹¹, moths¹² and ants¹³, among others. These numbers are somehow inversely related to sucrose strength given the convention adopted, i.e., a bee with a SRS of 1 responded only to the highest concentration but not to the lower ones. A bee with a SRS of 6 responded to all 6 sucrose concentrations, including the more diluted ones. We have specified this in the Methods section (lines 325-328).

4. In rows 218-220 the authors state that: "While the participation of OA in the enhancement of appetitive responses was predictable based on prior results the finding that DA has a similar effect represents a novelty." While it is true that there are indications from other works that although OA and DA are the main regulators of appetitive and aversive learning,

respectively, in insects they are not restricted to only one kind of learning but can also modulate the opposite one (e.g. Klappenbach et al 2013). This could also be part of the discussion.

- The Reviewer is right in mentioning this possibility. We have therefore included it in our new version and cited the paper by Klappenbach (see lines 242- 248).

5. In the introduction and discussion the findings could be more related to the wider context of how motivational states can affect learning and memory not just in invertebrates but other species including humans.

- We thank the Reviewer for this suggestion. As mentioned in our answers to Reviewer 1, the length constraints of the journal impose concision and render very difficult to elaborate in a detailed way on broad topics such as the one mentioned by the Reviewer. We have tried, nevertheless, to highlight the importance of motivational processes for learning and memory, which helped clarifying our study goals. We explained that motivation is central to animal and human behavior as it affects decision-making processes, event searching or avoidance, and consequently, what the underlying nervous systems learns and remembers (lines 58-62). If besides providing specific messages to conspecifics (the function for which they have been selected), pheromones change an animal's motivation, they will exert important consequences on its capacity to learn and memorize. Works on animal and human learning related to motivational states have been cited to support our arguments.

6. The expression "in a durable way" is used in row 52 and 72 but it is not entirely clear what the authors mean by this.

- We thank the Reviewer for pointing out this unclear expression. We have modified the wording to indicate that we asked if pheromone exposure induces persistent motivational changes affecting the way in which animals learn and memorize cues when the pheromones are no longer present (lines 57-58; see also lines 84-85).

7. In row 82 "stating" should be starting.

- Corrected. Thank you for spotting the typo (line 94).

8. In row 103 instead of "memory formation" it may be more appropriate to use "memory retention".

- We have replaced the term as suggested (line 118).

9. in Supplementary Table 1 the signs should be arranged more clearly showing the corresponding signs in all the rubrics.

- Modified as requested.

Reviewer #3 (Remarks to the Author):

The authors have asked a fascinating question, if pheromones, chemical messengers within an species, can influence cognitive processes. I think that the results are clearly presented and the experiments are appropriately designed and conducted. I believe that the effects are real, but I have some concerns about calling 2-heptanone and geraniol pheromones.

- We thank the Reviewer for her/his overall positive appreciation of our work. We have tried to answer her/his concerns below.

2H is a component of mandibular alarm pheromone, but there is lack of good evidence that it has an alarm function. For example, the Shearer and Boch (1965) paper showed repellent effects (bees avoided alighting at the nest entrance) but did not report the amount of the compound that they used. In Boch and Shearer (1971), which is a better reference because it provides more quantitative details, they explain that they provided $14 \pm 2 \mu\text{l}$ of the test substance (they tested multiple ones, including 2-H) onto a small cork, as described in their 1965 paper. However, this is not a realistic concentration, and the aversion could be due to simply the very high amount that this represents. Papachristoforou et al. (2012) reported that each honey bee had an average of $0.0386 \mu\text{l}$ of 2-H. If this true, then Boch and Shearer (1971) tested 362 bee equivalents of 2-H. If we look at Papachristoforou et al. (2012), the function of 2-H is not alarm but rather a toxin injected into the bodies of hive parasites. In this case, I suppose it could serve as a warning message to nestmates who smelled it, but Papachristoforou et al. (2012) did not demonstrate this.

There is much stronger evidence for isopentyl acetate (IPA) as an alarm pheromone component, and I imagine that the authors would achieve a similar effect with this compound. However, I would still caution them about the use of the term “pheromone”, which typically refers to a highly specific blend of semiochemicals for which the exact proportions are usually highly relevant.

Similarly, although GER is a component of the Nasanov gland, which produces a blend of compounds that are a true pheromone, it is unclear if GER is, by itself, a true pheromone. GER is a common volatile in the headspaces of flowering plants that are visited by bees for nectar and pollen. Given that the authors have demonstrated that the effect of GER is to enhance appetitive motivation, there seems no need to describe GER as a pheromone. Rather, it a common odor that is associated with food.

- We thank the Reviewer for mentioning these several points that we have separated in three main topics to facilitate our answers: 1) the role of 2-heptanone; 2) the use of the term pheromone; 3) the role of GER.
- The first point raised by the Reviewer refers to the role of 2H. The Reviewer affirms that there is a lack of evidence concerning the role of this substance as alarm pheromone. The Reviewer underlines the possible role of 2H as a toxin when injected through biting *via* the mandibular glands, highlighting the work of Papachristoforou et al.¹⁴. Although this work has indeed shown that bees inject 2H and paralyze thereby potential enemies, **this finding does not invalidate the possibility that 2H act also as**

an alarm pheromone. Arguing the opposite would be as extreme (and logically flawed) as stating that because pheromone components modulate learning and memory, this is all what they do, and their role as chemical messengers should be ignored.

In the major compendium existing on bee pheromones so far¹⁵, John B. Free places 2H among the alarm pheromones (*Chapter 14: Alarm and Aggression Pheromones*). He reviews therein several arguments – some of them mentioned by the Reviewer – that raise doubts about the role of 2H as alarm substance. The senior author of this article (MG) has also reviewed evidence in favor and against this view, including the role of 2H as a toxin¹⁶ so that these facts will not be repeated here. That review¹⁶ acknowledged that 2H might not directly participate in the coordinated attack of defending bees; yet, **it highlighted the use of this substance in various aversive contexts.** Free himself wrote as a conclusion on 2H:

“However, despite the conjectures that have been made [on the role of 2H] it is doubtful whether 2H-heptanone serves a major purpose other than guiding defending bees to a target.”¹⁵

In the context of the present work, we focused on the aversive valence of 2H. This was the main reason for using this substance, irrespective of its specific role in a coordinated defensive response. As mentioned above, the role of 2H could be debated but its use to signal aversive events seems out of question.

We have therefore modified our wording, excluding mentions to the role of 2H in an alarm context, but insisting on its negative valence and use in relation to aversive events (see lines 77-78, 273-275).

- The second point raised by the Reviewer is a criticism of our use of the term “pheromone” as it refers to a blend of substances, occurring in a specific ratio. The argument behind is that given this characteristic composition, we should not use the term “pheromone” in our work as we used of pheromone components instead of complete blends.

In this point, we agree – yet partially – with the Reviewer.

We agree on the fact that mostly (yet not always) pheromones are multicomponent blends, and on the fact that we used single components. This is unquestionable.

The argument applies to GER, which we used because it is the major component of the attractive pheromone of the Nasanov gland (which has at least 6 other minor components¹⁷). The real pheromone is difficult to obtain commercially as synthetic formulations are generally simplified versions with 2 or 3 components. A common use is, therefore, to use GER as a partial replacement of the Nasanov pheromone because it is a highly attractive component of the pheromone in tests in which components are tested separately. This is the strategy adopted in our work.

We acknowledge nevertheless that, strictly speaking, we should specify that GER is a pheromone component and not a pheromone.

The same arguments do not apply, however, to 2H, which is a single-component pheromone, at least in the way it has been historically described¹⁵.

In spite of this difference, we have adopted the convention of speaking about “pheromone components” instead of “pheromones”, when appropriate, and changed accordingly the title of our manuscript and the multiple mentions to pheromones in our text. We thank the reviewer for pointing out this inconsistency.

- The third point raised by the Reviewer refers to the presence of GER in some flower bouquets. This argument is used by the Reviewer to minimize the possible role of GER as pheromone in our experimental context.

This argument is fallacious, as the Reviewer probably knows that a vast majority of pheromone components appear in plant odors, **without this questioning their role as pheromone components**. Molecules integrating pheromones are commonly found in nature and are shared by many plant species. This may be related to the fact that the number of potentially small molecules that can be created, and that are chemically stable and non-toxic, is likely limited. Thus, arguing that we should not call a pheromone component as such because it appears in floral odors is inappropriate.

The Reviewer herself/himself is victim of his/her own argument as she/he suggested that we should use IPA (isopentyl acetate) rather than 2H because “There is much stronger evidence for isopentyl acetate (IPA) as an alarm pheromone component”. In fact, IPA is also present in the fragrance of *Rosa centifolia*¹⁸ and in several species of *Narcissus*¹⁹ regularly visited by honey bees. Thus, it is also a floral component found in dozens of floral species (see <http://www.pherobase.com/database/floral-compounds/floral-taxa-compounds-detail-isopentyl%20acetate.php>).

Considering the concentration of GER offered in our work, well above the one found in flowers (in ppm), we maintain that GER delivered in our work acted as pheromone component with positive valence, consistent with its participation in the pheromone of the Nasanov gland. **We have added a discussion on this point in lines 267-277.**

I think that the manuscript is quite interesting and still worthy of publication, but I suggest that the authors reduce their claims about these being pheromones. At best, they are pheromone components and for GER it not clear if the effect is due to GER being an odor associated with food or a component of Nasanov pheromone (but not the entire pheromone itself). I understand that making the claim that these are pheromones increases the potential impact of the study, but the fact that certain odor compounds can alter appetitive motivation in bees is a simpler claim that more closely follows what the authors have found.

- As explained above we have followed the Reviewer’s advice and spoke about pheromone components when appropriate. We thank the Reviewer for this suggestion.

A more minor question is why the pre-exposure interval of 15 min chosen? Is there a biological basis for this?

- Previous experiments done in our group using pheromone components such as IPA showed that this exposure time was enough to trigger different physiological effects and consistent behavioral changes²⁰⁻²². We thus adopted this exposure time hypothesizing that the effects induced by pheromones during this period could also translate into learning and memory capacities. This is specified in the Methods section (lines 303-307).

References

- 1 Szyszka, P., Gerkin, R. C., Galizia, C. G. & Smith, B. H. High-speed odor transduction and pulse tracking by insect olfactory receptor neurons. *Proceedings of the National Academy of Sciences* **111**, 16925-16930 (2014).
- 2 Pannunzi, M. & Nowotny, T. Odor Stimuli: Not Just Chemical Identity. *Frontiers in Physiology* **10** (2019).
- 3 Rouyar, A. *et al.* Unexpected plant odor responses in a moth pheromone system. *Frontiers in Physiology* **6** (2015).
- 4 Scheiner, R., Page, R. E. & Erber, J. The effects of genotype, foraging role, and sucrose responsiveness on the tactile learning performance of honey bees. *Neurobiol Learn Mem* **76**, 138-150 (2001).
- 5 Scheiner, R., Page, R. E. & Erber, J. Responsiveness to sucrose affects tactile and olfactory learning in preforaging honey bees of two genetic strains. *Behav Brain Res* **120**, 67-73 (2001).
- 6 Scheiner, R., Pluckhahn, S., Oney, B., Blenau, W. & Erber, J. Behavioural pharmacology of octopamine, tyramine and dopamine in honey bees. *Behav Brain Res* **136**, 545-553 (2002).
- 7 Scheiner, R., Barnert, M. & Erber, J. Variation in water and sucrose responsiveness during the foraging season affects proboscis extension learning in honey bees. *Apidologie* **34**, 67-72 (2003).
- 8 Scheiner, R., Page, R. E. & Erber, J. Sucrose responsiveness and behavioral plasticity in honey bees (*Apis mellifera*). *Apidologie* **35**, 133-142 (2004).
- 9 Erber, J., Hoormann, J. & Scheiner, R. Phototactic behaviour correlates with gustatory responsiveness in honey bees (*Apis mellifera* L.). *Behav Brain Res* **174**, 174-180 (2006).
- 10 Scheiner, R. & Arnold, G. Effects of patriline on gustatory responsiveness and olfactory learning in honey bees. *Apidologie* **41**, 29-37 (2010).
- 11 Belay, A. T. *et al.* The foraging gene of *Drosophila melanogaster*: Spatial-expression analysis and sucrose responsiveness. *Journal of Comparative Neurology* **504**, 570-582 (2007).
- 12 Hostachy, C. *et al.* Responsiveness to sugar solutions in the moth *Agrotis ipsilon*: parameters affecting proboscis extension. *Frontiers in Physiology* **10** (2019).
- 13 Perez, M., Rolland, U., Giurfa, M. & d'Ettorre, P. Sucrose responsiveness, learning success, and task specialization in ants. *Learn Mem* **20**, 417-420 (2013).
- 14 Papachristoforou, A. *et al.* The bite of the honeybee: 2-heptanone secreted from honeybee mandibles during a bite acts as a local anaesthetic in insects and mammals. *PLoS ONE* **7**, e47432 (2012).
- 15 Free, J. B. *Pheromones of Social Bees*. (Comstock Publishing Associates, 1987).
- 16 Nouvian, M., Reinhard, J. & Giurfa, M. The defensive response of the honeybee *Apis mellifera*. *J Exp Biol* **219**, 3505-3517 (2016).
- 17 Pickett, J. A., Williams, I. H., Martin, A. P. & Smith, M. C. Nasonov pheromone of the honeybee, *Apis mellifera* L. (Hymenoptera: Apidae). Part 1. Chemical characterization. *J Chem Ecol* **6**, 425-434 (1980).
- 18 Brunke, E.-J., Hammerschmidt, F.-J. & Schmaus, G. Scent of roses - recent results. *Flav. Fragr. J.* **7**, 195-198 (1992).
- 19 Dobson, H. E. M., Arroyo, J., Bergström, G. & Groth, I. Interspecific variation in floral fragrances within the genus *Narcissus* (Amaryllidaceae). *Biochem. Syst. Ecol.* **25**, 685-706 (1997).
- 20 Urlacher, E., Frances, B., Giurfa, M. & Devaud, J. M. An alarm pheromone modulates appetitive olfactory learning in the honeybee (*Apis mellifera*). *Front Behav Neurosci* **4** (2010).

- 21 Baracchi, D., Devaud, J. M., d'Ettorre, P. & Giurfa, M. Pheromones modulate reward responsiveness and non-associative learning in honey bees. *Sci Rep* **7**, 9875 (2017).
- 22 Rossi, N., d'Ettorre, P. & Giurfa, M. Pheromones modulate responsiveness to a noxious stimulus in honey bees. *J Exp Biol* **221** (2018).

REVIEWERS' COMMENTS:

Reviewer #1 (Remarks to the Author):

The authors have addressed my concerns and I think this manuscript is looking great, and ready for publication.

Reviewer #2 (Remarks to the Author):

The manuscript has been significantly improved and the authors addressed all the issues raised concerning the original submission. I fully support the publication of the paper in this form.

Reviewer #3 (Remarks to the Author):

I have reviewed the responses to my original comments and I am satisfied by the authors' responses, particularly, the change from pheromone to pheromone components and the inclusion in the Discussion of the questioned role of 2H as a pheromone component and the potential role of GER in floral scents. I recommend publication.